# Structural Biology for the Molecular Insight between Aptamers and Target Proteins

**DOI:** 10.3390/ijms22084093

**Published:** 2021-04-15

**Authors:** Ning Zhang, Zihao Chen, Dingdong Liu, Hewen Jiang, Zong-Kang Zhang, Aiping Lu, Bao-Ting Zhang, Yuanyuan Yu, Ge Zhang

**Affiliations:** 1School of Chinese Medicine, The Chinese University of Hong Kong, Hong Kong, China; zhangningning@link.cuhk.edu.hk (N.Z.); zihaochen@cuhk.edu.hk (Z.C.); hewenjiang@cuhk.edu.hk (H.J.); maxzhangzk@cuhk.edu.hk (Z.-K.Z.); 2Guangdong-Hong Kong Macao Greater Bay Area International Research Platform for Aptamer-Based Translational Medicine and Drug Discovery, Hong Kong, China; crawling1994@163.com (D.L.); aipinglu@hkbu.edu.hk (A.L.); 3Law Sau Fai Institute for Advancing Translational Medicine in Bone & Joint Diseases, School of Chinese Medicine, Hong Kong Baptist University, Hong Kong, China; 4Institute of Integrated Bioinformedicine and Translational Science, School of Chinese Medicine, Hong Kong Baptist University, Hong Kong, China

**Keywords:** aptamer, structure, interaction feature, binding affinity, modification strategy

## Abstract

Aptamers are promising therapeutic and diagnostic agents for various diseases due to their high affinity and specificity against target proteins. Structural determination in combination with multiple biochemical and biophysical methods could help to explore the interacting mechanism between aptamers and their targets. Regrettably, structural studies for aptamer–target interactions are still the bottleneck in this field, which are facing various difficulties. In this review, we first reviewed the methods for resolving structures of aptamer–protein complexes and for analyzing the interactions between aptamers and target proteins. We summarized the general features of the interacting nucleotides and residues involved in the interactions between aptamers and proteins. Challenges and perspectives in current methodologies were discussed. Approaches for determining the binding affinity between aptamers and target proteins as well as modification strategies for stabilizing the binding affinity of aptamers to target proteins were also reviewed. The review could help to understand how aptamers interact with their targets and how alterations such as chemical modifications in the structures affect the affinity and function of aptamers, which could facilitate the optimization and translation of aptamers-based theranostics.

## 1. Introduction

Aptamers are single stranded RNA or DNA oligonucleotides identified by selection against specific target molecules in vitro from large random sequence libraries (termed systematic evolution of ligands by exponential enrichment, SELEX) [1,2]. They are considered promising therapeutic agents against multiple diseases due to their high affinity and specificity towards target proteins. Up to now, among the 21 aptamers that are under clinical trials for safety and efficacy evaluations against various diseases, 16 of them have progressed towards final-phase development [3], and two of them are under promising phase Ⅲ clinical trials, while the clinical trials of the remaining three aptamers were terminated or unknown so far [3]. Even though numerous protein–aptamer complexes were investigated from the perspective of binding affinities, we still have little knowledge for the structural information of these complexes. The structural information could help researchers to understand the interactions between aptamers and their target proteins and, therefore, could help to explore the corresponding modification strategies for optimizing the affinity and efficacy of the aptamers.

There are some challenges in resolving 3D structures of aptamer–protein complexes. The structure of the short unbounded aptamer in solution can be determined by nuclear magnetic resonance (NMR) [4]. However, the structural heterogeneity of the aptamer in solution is a big concern. Furthermore, sizes of aptamer–protein complexes are usually too big to be solved by NMR. Crystallization of the binding complex and structural determination by X-ray crystallography could be helpful but are rather challenging. In view of this, the combination of both NMR and X-ray crystallography could be feasible when aptamers undergo conformational rearrangement during interactions with proteins [5]. The structural determination for aptamers and target proteins, in most cases, could provide a blueprint for understanding the specificity of the interaction and therefore provide information for the optimization of the interaction through modifications or nucleotide substitutions on aptamers. In this review, we reviewed the methods for resolving structures of aptamer–protein complexes and for analyzing the interactions between aptamers and target proteins. We summarized the general features of the interacting nucleotides and residues that were involved in the interactions between aptamers and proteins. Approaches for determining the binding affinity between aptamers and target proteins as well as modification strategies for stabilizing the binding affinity of aptamers to target proteins were also reviewed. We focus on recent research developments of structural studies between aptamer and target proteins, with critical insights into the features of aptamer–protein complexes and remaining challenges. This could not only facilitate the translation of the aptamer-based theranostics, but also provide a basis for the virtual screening of aptamers in future.

## 2. Progress in Structure Determination for Aptamer–Protein Complexes by NMR Spectroscopy, X-ray Crystallography and Cryo-EM

The three most common methods used to study the three-dimensional structures of biological macromolecules are X-ray crystallography, NMR spectroscopy and cryo-electron microscopy (cryo-EM). Each approach owns superiorities and limitations (Table 1). Because of overlapped spectrum and signals, the molecular weight of the biomolecules that have been determined by NMR spectroscopy is limited to approximately 30–40 kDa. Additionally, X-ray crystallography can be employed to analyze structures of averagely large molecules only in the case of the crystals could provide suitable quality of diffraction data. Intensive work is necessary to generate crystals that produce well-diffracting data. While crystallization of the sample is not demanded in cryo-EM studies, which is an amazing advantage over crystallography when crystallization is hindered by the dynamics of the target protein. Before analyzing the data and calculating the three-dimensional structure, good samples are necessary to obtain good data.

### 2.1. Structural Determination by NMR

When NMR technique was employed, the addition of specific ligands was necessary to ensure a stable aptamer conformation. A previously well-established approach has created the sequential assignment of NMR signals using ^13^C/^15^N labelled protein and triple-resonance experiments [6]. The integrated process has been profoundly computerized and automated, from signal assignment to structure determination. Nonetheless, analyzing nucleic acid structure by the means of ^13^C/^15^N-labelled nucleic acid and heteronuclear multidimensional NMR technique, which has been developed in the beginning of 1990s, is still under development and needs improvement [7]. Accordingly, in the 1990s, NMR spectroscopy was utilized to calculate the aptamer structures with the addition of a variety of supplements including the cofactors ATP [8,9] and FMN [10], the amino acids L-arginine and L-citrulline [11] and the aminoglycoside antibiotic tobramycin [12]. Particularly, in terms of RNA structures, NMR spectroscopy successfully facilitated the determination of these structures by fixing the RNA conformations in defined compressed conformations with high-affinity ligands and additives, resulting in analyzable narrow NMR signals.

Structure determination for aptamers longer than 50 nt demands non-uniform isotopic labeling strategies. A synthetically straightforward approach, called a divide and conquer approach, has been applied to study large RNAs where spectral overlap and fast relaxation are limiting [7]. The three-dimensional structure of the whole RNA was generated by spectral assignments and experimental constraints of the smaller fragments. Whenever the length of the aptamer surpasses 50 nt, it becomes challenging to process spectral overlaps in the sugar region and troublesome to make coherence preservation due to the increasing proton and carbon linewidths of slow asymmetric tumbling (leading to efficient the ^1^H-^13^C dipolar relaxation). In addition, during in vitro enzymatic transcription, less complicated NMR spectra is created by selective base labeling according to the type of the residue or the resonance type, which merely requires the blend of labeled nucleotide triphosphates (NTPs)—prepared by chemical or chemi-enzymatic approaches— and the remaining unlabeled NTPs [13,14,15]. The most commonplace method to simplify the spectrum of an RNA aptamer is an alternative strategy named segmental labeling, which does not influence the structure of the biomacromolecule. The approach is especially required when there are interactions between long-distance residues of an RNA that make divide and conquer impossible. As a consequence, the method decreases the amount of overlapping resonances in heteronuclear NMR experiments [16]. Taken together, the above approaches allow the most possibility to obtain good samples that yield unambiguous data.

### 2.2. Structural Determination by X-ray

X-ray crystallography is a technique applied to precisely determine the three-dimensional structure of a biological macro-molecule. The proficiency to create a three-dimensional crystal of a greatly purified protein or nucleic acid for diffraction is strongly necessary by X-rays [17]. The most general method used for crystallization is vapor diffusion in which a small volume (~0.5–10 μL) of a concentrated solution of aptamer–protein complex is combined with a precipitant solution, and then to equilibrate against a reservoir that contains a higher concentration of the precipitant, called the mother liquor [18]. As the protein amount used for testing of each condition is small, this technique allows for the screening of a much broader spectrum of conditions. Therefore, numerous variations of commercial precipitants could be screened to expect crystals. If obtaining an initial crystal, multiple optimization steps are generally engaged in to improve crystal quality, including fine screening for protein concentration, precipitant concentration, pH, salt concentration and incubation temperature. Diffraction-quality crystals in cryobuffer are immediately conserved in liquid nitrogen. Programs such as HKL2000 among others are used for collection of diffraction data, and then multiple strategies are applied to obtain initial phase that can be utilized to calculate a crystal structure [17] including molecular replacement (MR), multiple isomorphous replacement (MIR) and anomalous dispersion (MAD).

When the technology was applied in the structural determination of aptamer–protein complex, the most crucial step in the whole process was the acquisition of the diffraction-quality crystals. High purity and high concentration of the aptamer–protein sample are the first precondition of a desirable crystal. Additionally, commercial precipitants are frequently tested in the first-round screening to narrow the range of further conditions of buffers used for crystallization. Recently, Christina et al. determined a complex of Ghrelin and its L-aptamer NOX-B11 by optimizing crystallization condition using multiple cycles of micro seeding [19]. A single crystal exhibited high-quality X-ray diffraction data to a resolution of 2.65 Å, and the result also depicted that because NOX-B11 was only capable of forming a complex with the octanoylated form, thus it interacts with the fatty acid usually containing hydrophobic N-terminus [19].

### 2.3. Structural Determination by Cryo-Electron Microscopy (Cryo-EM)

When determining the structure of biological macromolecules and assemblies, researchers prefer to choose the biophysical technique cryo-EM for non-crystalline single particles. For decades, its application potential in drug discovery has been heavily limited by two cruxes: the minimum molecular weight of the structures it can be used to research and the resolution of the images. Nevertheless, recent technological advances are revolutionizing the applicability of cryo-EM, including the improvement of direct electron detectors and more powerful computational image analysis techniques, spurring an explosion in the number of high-resolution structures of sizable macromolecular assemblies. When more and more “intractable” targets that were previously not available to X-ray crystallographic analysis have been currently determined by the single-particle cryo-EM, the technique has been expected to play a promisingly important role in drug discovery, as it is making an advancement [20]. Since the value of structural information derived from X-ray crystallography and NMR spectroscopy is thoroughly established today, rapid advances in cryo-EM have raised a burst of interest in the potential application of the technique in the area of drug discovery. 

Multiple successive steps are engaged in determining a structure by cryo-EM, including sample handling, EM grid establishment, image acquisition and image processing. Due to the specialized improvements at all these steps in the last decade, the potential to apply cryo-EM in drug discovery has been extensively enhanced. Three main aspects of these achievements are acquiring higher-resolution cryo-EM maps, managing smaller proteins and increasing the throughput [20].

The structure-specific endonuclease XPF-ERCC1 participates in several DNA damage repair pathways including nucleotide excision repair (NER) and inter-strand crosslink repair (ICLR) [21,22]. Morgan et al. reported cryo-electron microscopy structures of both DNA-free and DNA-bound human XPF-ERCC1. The structure indicated that DNA-free XPF-ERCC1 undergoes an auto-inhibited conformation in which the ERCC1 (HhH)2 domain is shielded and not exposed to the XPF catalytic site due to the XPF helical domain [23]. DNA connection engagement discharges the ERCC1 (HhH)2 domain to be able to pair with the XPF-ERCC1 nuclease/nuclease-like domains. The structure solved by cryo-EM facilitated explaining how DNA junction substrates catalytically stimulated the function of XPF-ERCC1 [23].

### 2.4. Other Methods for Structural Determination

Circular dichroism (CD) spectroscopy is an elementary device for the characterization of G-quadruplex (G4) structures of aptamers [24]. A recent chemometric analysis of nucleic acid CD spectra was applied to categorize nucleic acid structures using a library of sequences and three-dimensional structures that broadened the scope of topologies to include multi-stranded triplex and quadruplex forms [25]. Another powerful tool was native mass spectrometry (MS), which has evolved as a robust technique for direct probing of native state proteins and non-covalent protein complexes under near physiological conditions [26]. Rich structural biology information can be acquired from native MS experiments in terms of protein complex stoichiometry [27,28], protein–protein interaction interfaces [29] and protein–lipid interactions [30]. Therefore, MS could be an alternative strategy to study the structural characteristics of the protein–aptamer complex.

## 3. Interfaces between Aptamer–Protein Complexes

### 3.1. Structures of Aptamer–Protein Complex in Database

Novoseltseva et al. have reviewed totally forty-five structures of aptamer–protein complex in the dataset in 2018 [31]. By April 2021, this number had increased to seventy-three in the Protein Data Bank (PDB). All these complex structures were solved by X-ray crystallography with the resolution between the range of 1.8–4.5 Å. We examined some of them in this review by summarizing four structural and affinity characteristics of aptamer–protein complexes, including polar contacts number, A^2^—interface area, aK_D_—apparent dissociation constant and ΔGb—Gibbs free energy change during complex formation (Table 2).

The polar contacts include hydrogen bonds and electrostatic interactions. The parameters of the aptamer–protein complexes display considerable variation. For instance, the interface area ranges from 410 [65] to 2088 Å^2^ [36], while the binding affinities between each pair of these two aptamer–protein complexes have no obvious differences. Dissociation constant (K_d_) represents the binding strength between two molecules. Basically, the smaller the value, the stronger the interaction is. Among all the published K_d_ values between the aptamer and its target protein, the constants fluctuate mildly around the levels of pm-nm, suggesting quite a strong binding affinity between each pair. Another fundamental characteristic of binding affinity is the Gibbs free energy change in the complex formation ΔGb. The values were measured for the complexes with accepted obvious dissociation constants of the given temperature. Despite the obvious differences in the polar contacts and interface area of each complex, they own stable three-dimensional structures and considerably strong affinity between each pair. Accordingly, the methods of structural determination and approaches of interaction research are reviewed in the subsequent parts.

The NMR method was also utilized to study the binding between aptamer and its target, but not for complex structural determination directly. In the case of an IgG aptamer, determining the binding surface of the Fc fragment was achieved by the analysis of the chemical shift perturbation of the IgG Fc fragment upon aptamer binding [42]. The ^1^H-^15^N HSQC spectra of the ^15^N-labelled Fc fragment showed that some signals were perturbed noticeably upon aptamer binding, which exhibited the perturbed residues in the binding interface, confirmed by X-ray crystallography [43].

Alterations in imino proton signals are frequently used in interaction analyses. In the case of a spermine-binding aptamer, the imino proton spectra showed obvious disparities in the absence and presence of spermine [66]. Although the spectrum was not influenced by the addition of Mg^2+^, a noticeable change was monitored before and after the addition of spermine. The extended and vanished signals of the spectrum of the terminal stem indicated that the base pairs of the domain opened upon binding of spermine. Another noticeable change was observed in TOCSY spectra that there were perturbations of pyrimidine H5–H6 signals upon spermine binding, suggesting a conformational change in the molecule. Claiming the tertiary structure remains to be calculated, these chemical shift changes can be temporarily mapped onto the secondary structure of the aptamer. When it comes to the investigation of the structure, dynamics and interaction of proteins and nucleic acids in living cells, a modified NMR technique, in-cell NMR, has been advanced to collect the above information [67].

The interaction force, which contributes to the complex formation, has been investigated in detail, among which there are several common features. The first characteristic is named cation–π interactions that build up the clear-cut protein–DNA or protein–RNA recognition, which customarily occurred between nucleic acid bases and amino acid residues, especially with the residues that carry full or partial positive charges (Arg, Lys, Asn, Gln) [68]. While the second characteristic is named π–π interactions that likewise take place between nucleic acid bases and Arg residues [68,69,70] (Figure 1). In spontaneously present protein–nucleic acid complexes, both cation–π and π–stair motifs are popular [68,70,71]. Particularly, in the interfaces of the protein–nucleic acid complexes that conceal proportionately large surface areas, clustered cation–π and π–stair motifs make it remarkably fitting for the construction of these extremely complementary, interdigitated interfaces. Compared to amino acid residues Gln and Asn that occasionally participate in cation–π interactions, positively charged Arg and Lys appear much more repeatedly in cation–π interaction motifs [69,70]. Thereupon, positively charged protein surfaces are normally favored during in vitro aptamer selection, especially those with clustered elemental residues.

### 3.2. The Features of Aptamers That Contribute to Binding

According to some typical features of the nucleic acid bases that are involved in the interface, some examples exhibited the common ones. Gunaratne et al. determined the three-dimensional structure of a protein–aptamer complex at 2.0 Å resolution by adopting the method of molecular replacement. The tertiary fold of the aptamer, 11F7t, displays an enlarged molecular surface for interactions with the protein, GD-FXaSer195Ala, with approximately 1400 Å^2^ of solvent-accessible surface area buried inside the protein–aptamer complex. The contact force function primarily in the complex was specified by 15 hydrogen bonds (Figure 2a). The nucleotide bases in a central loop encompassing C8, A10, A21, C28–C30 and the proteinase domain residues including Leu59, Arg64, Val88, Iso89, Asn92, Arg93, Lys236 and Arg240 [71] contributed mainly to the interaction between GD-FXaSer195Ala and 11F7t. Another work showed that the major groove of the aptamer was bound to the arginine-rich helix domain of the target protein, Rev (Figure 2b), ending up with several contacts additional to those formed by the wild-type RNA. Once bound with the target protein, the aptamer would hamper Rev oligomerization at the interface [72].

### 3.3. The Features of Proteins That Contribute to Binding

In the binding interfaces of the protein–aptamer complexes, apart from stacking interactions engaging the bases, protein epitopes that interact with aptamers are substantially electropositive and hydrogen bonds and charge–charge forces predominantly contribute to the interaction between the aptamer and the target protein. In terms of the number of all hydrogen bonds and charge–charge interactions, it multiplies approximately proportionately with the interface area for the typical aptamers [73].

The crystal structure of the VWF A1-ARC1172 complex presents that the aptamer embraces a typical three-stem structure of principally B-form DNA, including three noncanonical base pairs and nine individual residues, six of which maintain the balanced structure through the agency of base–base or base–deoxyribose stacking interactions. Classic cation–π interactions involving Arg, Lys and Gln residues characterize the aptamer–protein interface (Figure 2c), which is stabilized by H-bonds with neighboring bases [45]. 

An RNA aptamer r(GGAGGAGGAGGA) (R12) against bovine prion protein (bPrP) unveiled that both bPrP and its b-isoform could bind to R12 with high affinity [74]. Two quadruplexes constitute a dimer in virtue of the intermolecular hexad–hexad stacking. The binding sites between bPrP and R12 has been recognized as the two lysine clusters of bPrP, which interact with the exclusively organized phosphate groups of R12 to establish the electrostatic interaction that is basically responsible for the affinity of R12 to bPrP. Another force that may contribute to the affinity is the stacking interaction between the G:G:G:G tetrad planes and tryptophan residues. Two lysine cluster residues of one bPrP molecule is presumed to bind one R12 dimer synchronously, ending up with even higher affinity [74].

### 3.4. Conformational Changes upon Binding

The formation of aptamer–protein complex regularly occurs without a considerable change in the conformation of the protein. Compared to the protein in the unbound state, the protein in the complex mostly maintains the consistent conformation. However, there is an exception where the aptamer DNA might invoke an allosteric effect on HIV-1 reverse transcriptase [75]. Hydrogen–deuterium exchange analysis was used to explore the conformational changes in HIV-1 reverse transcriptase upon binding its aptamer DNA. When mapped onto the crystal structure of the complex, it was shown that the aptamer substantially rigidified the regions of the protein that were the most dynamic under unliganded pattern, even that which made no direct contacts with the nucleic acids.

The conformational changes in aptamers upon binding to target proteins remain poorly investigated. So far, there are only a few disclosed solution structures of pure aptamers that have corresponding co-crystal three-dimensional structures, e.g., the S8 aptamer [5] and the NFκB aptamer (Figure 3a,b) [76]. A significant conformational change is observed after the S8 aptamer binds to the target, exhibiting a configuration akin to that of the 16S rRNA sequence binding to S8. The individual RNA aptamer in solution adopts a helical structure with several non-canonical base pairs. With the help of an unusual combination of nucleobase interactions, binding of S8 results in a significant change in the RNA conformation that reestablishes the signature S8 recognition fold [5]. The structure of the protein-free RNA aptamer in solution is correctly folded and organized and exhibits imperceptible intermediate timescale dynamics. Swift NH solvent exchanges are powerful proofs to demonstrate that the G–C and A–U base pairs flanking the internal loop manifest increased solvent accessibility. While in the complex, the conformation of the RNA binding site core region has been substantially reshaped. In terms of the details, the base of G13 leapfrogs over U12 to marry with C28, which ends up disrupting the G13–A26 pair and creating a base triple through the minor groove edge of G11. The helix forces out the base of A26, while A26 still remains to stack beneath the surface of the adjacent U27 residue. A cluster of hydrogen bonds tethers a reassembled base quartet tightly, including the U12–U27 and A14–U25 base pairs and the functional groups of the dominant groove base edges. Residue A14 sticks to be a member of a Watson–Crick-type base pair; however, its mating partner alters from U25 to U27. Taken together, the above rearrangement of core nucleotide interactions appears quite exclusive among free or RNA-ligand complexes.

Conversely, the NFκB RNA aptamer adopts a pre-formed stem-loop structure that bears resemblance to natural B-form DNA binding partner. An induced fit takes place in an internal loop and the terminal tetraloop of the RNA upon binding the protein (Figure 3c). Pre-arranged characteristics are observed both in the RNA aptamer internal loop structure and in the complex, involving non-canonical base pairing and cross-strand base stacking. The free RNA aptamer structure is uncommonly endowed with a major groove that meticulously resembles B-form DNA rather than RNA. Upon protein binding, the internal loop and the terminal tetraloop of the RNA structure entangles and distort, respectively. Consequently, both pre-formed and induced fit binding interactions take place in the formation of protein–aptamer complex [76]. Besides, the aptamer pegaptanib, NX1838, has also exhibited induced fit that had been detected by NMR experiments [77]. Plenty of aptamers undergo conformational alterations upon binding to small molecules [78].

Taken together, during the process of aptamer–protein complex, the above cases suggest that compared to proteins, aptamers own a tendency to display stronger flexibility in accommodating the conformations that contribute to the stability of the complex.

### 3.5. Factors Contribute to the Stability of Aptamer Structure in Structure Determination

When resolving the structure, it is vital to maintain the stability of the aptamer. During the process of aptamer crystallization, usually a molecular chaperone is necessarily required. Koldobskaya et al. performed a fab antibody-assisted RNA crystallography to generate a binding site between the chaperone Fab BL3-6 and RNA by substituting four-nucleotide loop (UUCG) from the stable stem-loop region of the wild-type (WT) DIR2s aptamer with a 7-nucleotide GAAACAC motif [80]. Fab BL3–6 binds to the adjusted DIR2s aptamer with an affinity (Kd = 67 ± 20 nm) resembles the binding affinity reported for either Spinach RNA attached with GAAACAC motif [81] or isolated hairpin from class I ligase RNA [80]. Co-crystallization of DIR2s-OTB-SO3 RNA-fluorophore and Fab BL3–6 was achieved successfully, and the structure was finally determined at 1.8 Å resolution with one Fab-RNA–fluorophore complex molecule in each crystallographic asymmetric unit. 

While during the process of resolving structures of aptamers by NMR, researchers usually heated the aptamer before releasing the sample to the NMR spectroscopy equipment. Mashima et al. resolved an RNA 12-mer (R12) aptamer, r(GGAGGAGGAGGA), which was dissolved in a solution consisting of 100 mm KCl, 10 mm K-phosphate (pH 6.2) and 3 mm NaN_3_, with the final concentration as 1.0 mm. The sample was heated at 95 °C for 5 min, followed by continuous cooling to room temperature prior to the measurements [74]. Another group suspended the RNA samples in 0.35 mL 99.96% D_2_O or 90% H_2_O/10% D_2_O and annealed and contained 30–140 A_260_ OD units of RNA oligonucleotide (≈0.4–1.5 mm) [5]. The above solutions suggested that both the temperature and the concentration of the samples affect the stability of the aptamer structure in structure determination. 

## 4. Challenges in Structural Determination

### 4.1. Low Affinity of Aptamers to Proteins

Between the free state (I) and the binding efficient state (II) of the aptamer, an equilibrium occurs with a corresponding equilibrium constant K_E_. The binding competent state (II) is responsible for binding the target with a dissociation constant K_D_. Multiple factors can affect the binding affinity of the aptamers and their targets and their actual conformation in the formation of the aptamer–target complex, including salt concentration, pH value and the temperature of the sample. Not all aptamer configurations are capable of binding to their targets in a stable manner [82]. In some cases, the affinities of aptamers to targets after SELEX are not high enough, post-SELEX modifications would be employed on the interaction sites between aptamers and targets to improve the binding affinities. Structural determination could provide a molecular insight into the interaction characteristics of the aptamer–target complex and support clues for the nucleotides to be modified. Through the introduction of satisfactory chemical modifications, the aptamers in their binding efficient state (II) can be stabilized, and the affinity of the aptamers for their target ligands can be raised.

### 4.2. Methods for Determination of the Binding Affinity between Aptamers and Proteins

Structures of aptamer–protein complexes provide a profound insight into the interface between each other, whereas we need to quantify the binding strength between aptamers and target proteins by other biochemistry methods. Usually, different methods have distinct features and limitations (Table 3). Therefore, we might benefit from a better understanding of the methodology of the interaction research.

#### 4.2.1. Isothermal Titration Calorimetry (ITC)

Among the whole process of a biomolecular interaction, the quantity of discharged heat and consumed heat could be directly evaluated by a physical technique called ITC. The method works on the elemental principle of thermodynamics where connection between two molecules ends up with either heat generation or heat absorption, exothermic or endothermic, relying upon the pattern of binding [83]. One of the most remarkable features of the this analytical approach is that the ligand keeps in contact with a macromolecule under constant temperature [84]. ITC was used to investigate the thermodynamics of the interaction of thrombin binding aptamer (TBA) and modified TBA to thrombin [41]. To determine the value of stoichiometry (n), binding constant (K_b_) and binding enthalpy (ΔH), a prevalent model that presumes a single collection of tantamount binding sites was operated to fit in the calibrated data acquired from the experiment. The single set of the equivalent sites model is the most straightforward mathematical model to generate the value of K and ΔH. These thermodynamic parameters, including the binding Gibbs energy change, ΔG, and the entropy change, ΔS, are calculated from the equations ΔG = −RTlnK_b_ and TΔS = ΔH − ΔG, respectively. From the perspective of the thermodynamics, the variations of the values of the binding constants and the Gibbs energy suggest that the associations are greatly preferred under 25 °C.

#### 4.2.2. Surface Plasmon Resonance (SPR)

The basic principle of SPR functions via the cumulative fluctuation of conduction band electrons that are in resonance with the oscillating electric field of incident light, which will generate energetic plasmonic electrons by virtue of non-radiative excitation [85]. In the character of a multiplexed, surface-sensitive optical technique, SPR works based on the changes in the local refractive index to recognize and detect adsorption onto microarrays [86]. In order to create a set of balanced RNA microarrays, Li et al. have immobilized single-stranded DNA (ssDNA) onto the surface of the unmodified single-stranded RNA (ssRNA) through a covalent enzymatic ligation reaction [87]. With the help of surface ligation chemistry, whether a stable RNA microarray is or is not formed successfully is confirmed by SPR imaging (SPRI) measurements. SPRI measurements of microarrays have been broadly investigated for the research of DNA–protein, peptide–protein, protein–carbohydrate and protein–antibody interactions [88,89,90]. Li et al. confirmed that SPRI measurements of RNA aptamer microarrays can be adopted to investigate aptamer–protein interactions, and their RNA–DNA surface ligation chemistry has been managed to establish a five-component RNA microarray of potential aptamers for protein factor IXa (fIXa) [91]. Consequently, SPRI measurements ends up facilitating selecting the optimal aptamer for fIXa out of the total five RNA aptamer components [87].

#### 4.2.3. Atomic Force Microscopy (AFM)

From the molecular level, another promising technique named AFM is advantageous for detecting binding affinity and probing recognition properties [92]. A user-friendly way is provided by the technique to evaluate the molecular interaction between a ligand-functionalized AFM cap and a receptor-modified substrate through the detection of the unbinding events. Under physiological conditions, compared to other alternative sensitive methods employed for force calculation and biological macromolecular interaction research, AFM is more constructive in terms of precise force resolution, high spatial resolution and the operating efficiency [93,94,95]. AFM has been successfully applied in the measurement of the unambiguous interaction between the protein immunoglobulin E (IgE) and its selected 37-nt aptamer by Jiang et al. Poisson statistical formula was adopted to analyze the single-molecule unbinding force between IgE and the binding aptamer, as well as the individual unbinding force between IgE and its monoclonal antibody. Compared to the affinity between IgE and the antibody, the binding affinity between the aptamer and the target protein is relatively robust, which could match or even exceed that of the antibody to its antigen [92]. 

#### 4.2.4. Flow Cytometry

Flow cytometry is a technique that swiftly evaluates single cells or particles suspended in a salt-based buffer, as they stream past single or multiple lasers. Each particle is analyzed for visible light scatter and one or multiple fluorescence parameters [96]. It is a robust tool with applications in various disciplines, including immunology, virology, molecular biology, cancer biology and infectious disease monitoring. In the area of molecular biology, flow cytometry could be applied to detect the interactions between biomacromolecules [97]. Flow cytometry could be used for determining the binding ability of individual aptamers (fluorescently labeled) to live cells. Limitations of flow cytometry include that the instrument is relatively expensive and provides overwhelming information about the samples that may not always be necessary. Liang et al. have characterized the binding between aptamer candidates and several cell lines. They found that the aptamers they identified can specifically bind to rat osteoblasts but do not bind to other cells such as rat osteoclasts, rat peripheral blood mononuclear cells or rat liver cell line [98].

#### 4.2.5. Enzyme-Linked Oligonucleotide Assay (ELONA)

A variant of enzyme-linked immunosorbent assay (ELISA) for oligonucleotides and aptamers is commonly known as ELONA or aptamer linked immobilized sorbent assay (ALISA) [99]. It is essentially ELISA but the antibody is replaced by oligonucleotide or aptamer as the recognition molecule. The benefit of this method is that unlike antibody, aptamer can be easily labeled without significant effect on the affinity and specificity of the aptamer. In addition, aptamer can provide a constant source of high and uniform quality detection reagent. ELONA has been employed to measure the binding affinity and specificity of the full-length aptamer and one of its truncated versions for protein A that is expressed on the *Staphylococcus aureus* [99]. The full-length aptamer demonstrated significantly lower binding affinity and the 3-biotinylated variant resulted in a better K_d_ of 101.4 ± 5.9 nm than the 5-biotinylated variant with a K_d_ of 189.9 ± 13.0 nm, which could be possibly due to the assay conditions [100]. Sypabekova et al. have demonstrated the use of ELONA to study the specificity of ssDNA aptamers for MPT64 protein of *Mycobacterium tuberculosis* (Mtb) that causes tuberculosis (TB) [101].

#### 4.2.6. Surface-Enhanced Raman Spectroscopy (SERS)

SERS is a surface-sensitive approach that strengthens Raman scattering by molecules adsorbed on rugged metal surfaces or by nanostructures such as plasmonic–magnetic silica nanotubes [102]. Since Raman spectroscopy owns the advantages of both molecular specificity and high sensitivity due to the optical properties of plasmonic nanostructures, the application of SERS for biological and medical applications obtained much attention in the last decade [103]. In the process of SERS-based label-free detection, once the direct interaction between the samples and the SERS-based nanostructures occurs, vibrational spectroscopic information is obtained, thus providing essential fingerprint information of biological and biomedical samples with strengthened intensity [104]. Since aptamer–protein binding usually results in a conformational change in the aptamer structure, alterations in the SERS spectrum of the aptamer could be indicated to signal protein binding. Ochsenkühn et al. investigated that the structural change in the thrombin binding aptamer (TBA) on interaction with thrombin could be demonstrated by new peaks in the sensor spectrum owing to protein binding [105].

#### 4.2.7. Microscale Thermophoresis (MST)

Microscale thermophoresis is based on the conducted movement of molecules through temperature gradients, an effect named thermophoresis [106]. MST represents a powerful and convenient technique to quantify the binding affinities of protein–nucleic acid interactions. A spatial temperature difference ΔT results in a reduction in molecule concentration in the region of elevated temperature [107]. The size, charge and hydration shell of the molecules determine the characteristics of their directed movement along temperature gradients. At least one of the parameters mentioned above is changed when the binding of a ligand to a molecule occurs, ending up with different thermophoretic movements of the unbound and bound states [108]. Since aptamers can be easily acquired with all kinds of fluorescent dyes attached, they can be straightforwardly used in MST as the constant, fluorescent interaction partner [109]. Skouridou et al. adopted specific aptamer (T5) as a probe for testosterone (anabolic steroid) interactions [110]. With the application of MST technique, the specific aptamer (T5) displayed higher affinity to testosterone among other candidates. The dissociation constant evaluated with MST K_d_ = 5.7 nm was consistent with K_d_ values that were acquired from an apta-PCR affinity assay (K_d_ = 4 nm) [111]. 

#### 4.2.8. Bio-Layer Interferometry (BLI)

Bio-layer interferometry (BLI) is a label-free technique based on the real-time optical monitoring of biomolecular interactions [112,113]. Concisely, experiments are undertaken in classic multi-well plates containing a solution of one molecule in which a biosensor tip covalently functionalized with the second molecule is submerged. Tips are formed by a biocompatible layer to avoid non-specific interactions with the sensor and a thin layer covered with reactive groups. Once the functionalized biosensor is irradiated with a white light laser, interferometry variation occurring during association/dissociation steps is detected and the kinetic and thermodynamic parameters of the interaction is determined, with the corresponding precision of other physico-chemical approaches [114,115]. BLI has never been applied to investigate multivalent interactions due to its weak sensitivity to external factors such as aggregation and microfluidic troubles. When the technology was applied to aptamer–protein interaction, Wojciech et al. characterized the binding kinetics of the anti-recombinant human erythropoietin (rHuEPO) SOMAmers to rHuEPO, since it offers a high-throughput platform for determining the rate constants for SOMAmer–rHuEPO interactions [116]. The result demonstrated an alternative approach to generate high-affinity anti-rHuEPO aptamers [117] and contain distinctive nucleotide sequences from previously identified aptamers [116].

### 4.3. Modifications of Aptamers to Enhance the Stability and Binding Affinity of Aptamers

With the development of original aptamers, plenty of attempts and intensive efforts have been devoted to the modification of existing aptamers. The modification usually aims to essentially strengthen the insufficient stability of the aptamer, as well as to improve the affinity and selectivity of nucleic acid aptamers towards their target proteins. The classification of the modification strategies was briefly summed up in Table 4.

#### 4.3.1. Chemical Modification Strategies

Chemical modification was affirmatively the first and the most prevailing approach that we regarded for the optimization of aptamers. According to the specific position that has been modified on the aptamer, it included: (i) modifications on the inter-nucleotide linkages and (ii) modifications on the nucleotide base.

In terms of modifications on the inter-nucleotide linkages, it facilitates the stability of aptamer by protecting it from hydrolyzing by nucleases. In the case of a thrombin-binding aptamer (TBA), similar to many target-specific nucleic acid ligands, TBA possesses a noncanonical conformation in solution. The aptamer is more likely to generate an antiparallel two-tetrad G quadruplex (GQ) conformation in the existence of sodium, potassium or ammonium ions [118]. The process GQ formation is verified to be crucial for its binding with thrombin [119,120], thus modifications that diminish the GQ thermostability are unsatisfactory. Although loop modifications tend to insignificantly influences quadruplex thermostability, they usually impart heightened nuclease resistance to the aptamer [121,122]. Three TBA analogs with distinct loop modifications were synthesized by the researchers: the thiophosphoryl TBA analog (thio-TBA), the triazole-linked analog (triazole-TBA) and the analog possessing alpha-thymidine (alpha-TBA). To prevent oligonucleotides (ONs) from hydrolyzing by nucleases, inter-nucleotide modifications involving the thiophosphoryl modification and the triazole modification are well-known methods [123].

Apart from modifications on the inter-nucleotide linkages, another strategy regarding modifications on the nucleotide base was also prevalent. The establishment of anomeric nucleoside moieties (alpha-nucleosides) has also been confirmed to exhibit stronger enzymatic stability to ONs [124]. Adding hydrophobic groups to the nucleic acids is an effective tool to improve aptamer binding affinity towards their target proteins, resulting in the formation of additional hydrophobic interactions that contribute to the stability of the aptamer–protein complex. Besides, hydrophobic interaction, a stronger binding force than the conventional ones, also helps stabilize the nucleic acid scaffold. The chemical 5-(*N*-benzylcarboxamide)-20-deoxyuridine (Bn-dU) was used to modify the DNA aptamers against interleukin-6, nerve growth factor and platelet-derived growth factor BB (PDGF-BB) [48,53,55]. These modified aptamers are called slow off-rate modified aptamers (SOMAmers) with picomolar to nanomolar affinities against their targets. Take PDGF-BB and its SOMAmer SL5, for example, the modified nucleotides stabilized 5′ stem and hinge between domains and created extensive hydrophobic interactions in the SOMAmer–protein interface [48].

#### 4.3.2. Structural Modification Strategies

Apart from chemical modifications, another modification strategy called structural modification has been used to improve the stability of the aptamers and their affinity towards the target proteins. According to the structural features that have been modified on the aptamer, it included: (i) modifications on secondary structures, (ii) modifications on the interactive nucleotides and (iii) addition of structural groups.

To stabilize the secondary structures of DNA aptamers, the conformation of duplex flanks were added and also played an important role in modeling their in vivo surroundings [125]. Another example showed that structural modification leads to a significant enhancement in the binding affinity between an RNA aptamer and its target protein. As shown in the previous crystal structures, the two non-connected oxygen atoms of several phosphate groups located in the RNA backbone usually interact with amino acid side chains of the target protein through the medium of hydrogen bonds and/or salt bridges (Arg, Lys, His). Replacing the oxygen atoms by sulfur to yield a phosphorodithioate (PS2) linkage has been an effective tool to enhance the aptamer–protein affinity by the hydrophobic effect and/or strengthened polarizability of the PS2 moiety. A single PS2 substitution in AF83-1 (an aptamer against VEGF_165_) and AF113-1 (an aptamer against thrombin) gave rise to remarkable enhancements in affinity by about 1000-fold toward their cognate target proteins. Additionally, the crystal structure of PS2-modified aptamer–thrombin complex suggested that the PS2 moiety induced an altered but preferred interaction between the moiety and the neighboring amino acid side chains [58]. Di-thio-modified X-aptamers were the next generation of phosphorothioate aptamers, in which a C5 position of the nucleobase was modified with a drug-like performance [126]. An indole moiety and a methyl group were also used to strengthen the binding affinity of the TBA aptamer. Substituting the T at position 4 by 5-(indolyl-3-acetyl-3-amino-1-propenyl)-2-deoxyuridine (W) or 5-(methyl-3-acetyl-3-amino-1-propenyl)-2-deoxyuridine (K) afforded T4W and T4K, respectively, which improved the affinity compared to the unmodified aptamer TBA. As shown in the crystal structures, both T4W and T4K formed hydrophobic contacts with several hydrophobic thrombin side chains [60].

#### 4.3.3. Modified Aptamers in Clinical Trials

Despite aptamers promising alternatives to antibodies, only 21 of them are under clinical trials or used as drugs (Table 5). One aptamer, Macugen, has been approved for the treatment of age-related macular degeneration [127]. The virtual analysis of the induced-fit binding model was illustrated in Figure 4. The analysis showed that Macugen has undergone a considerable conformational change upon binding to the target protein, vascular endothelial growth factor (VEGF)165. In addition, most of the residues that contribute to the interaction between the aptamer and VEGF165 are positive-charged residues (R13, R14, K15, H16) (Figure 4b). Among the other aptamers, NU172 is currently in Phase II as an anticoagulant in heart disease treatments [3]. It blocks thrombin activity much more efficiently than TBA, the best-known thrombin binding aptamer. The crystal structure of the thrombin–NU172 complex uncovered that the aptamer undergoes a mixed duplex/quadruplex conformation whose backbone is a successive stacking of bases from the duplex to the quadruplex region [128]. Another aptamer, the first high-resolution crystal structure of a non-natural, mirror-image L-RNA aptamer binds to a natural L-protein. The L-protein is CCL2, a chemokine engaged in inflammatory processes that are dysregulated in various diseases [129]. PEGylation is one common modification method for aptamers in clinical trials to resist the renal filtration in vivo. The PEGylated L-aptamer (Emapticap pegol) has already been proved to be safe and well tolerated in several Phase I clinical trials in healthy volunteers. Additionally, this modified aptamer recently exhibited efficacy in diabetic nephropathy patients as well in a Phase IIa clinical study [3,130]. This is the first mirror-image oligonucleotide that was developed into clinical studies. More detailed molecular information of the target recognition of an L-aptamer finally establishes the basis for its effectiveness [54].

### 4.4. Heterogeneity of the Aptamers

Considering the tremendous potential of aptamers to fold into numerous tertiary structures, which was considered as “aptamers plasticity”, we may necessarily make sure that the aptamers were stable and folded correctly during the process of sample preparation. 

The preparation of aptamers for structural determination is relatively important. In detail, an example showed that the purification of an RNA aptamer that strongly binds transcription factor NF-B was achieved by denaturing 15% polyacrylamide gel electrophoresis. The aptamer was identified by UV absorbance and excised from the gel, refolded under the condition of 0.3 m sodium acetate and precipitated from ethanol [76]. An anion exchange column (MonoQ) was adopted to further purify the aptamer, and a gel filtration column was utilized to desalt the aptamer. The purified RNA sample was lyophilized, resuspended in water with the addition of 1 m NaOH to change to the solution condition of pH 6.8. To obtain the non-exchangeable proton data, the sample was further lyophilized to facilitate the preparation of a 99.99% D_2_O solution. Another example showed that aptamers selected against ribosomal protein S8 were synthesized in vitro with the help of T7 RNA polymerase and an artificial DNA template. The protocols of the preparation of unlabeled and isotopically labeled RNA molecules were described before [132]. The polyacrylamide gel electrophoresis (PAGE)-purified RNA molecules were dialyzed thoroughly against 10 mm KCl, 10 mm sodium cacodylate, under the condition of pH 6.6 and with the addition of 0.02 mm EDTA and finally lyophilized. The RNA samples were resuspended in 0.35 mL of 99.96% D_2_O or 90% H_2_O/10% D_2_O and annealed, and the final concentrations of RNA oligonucleotide was measured as 0.4–1.5 mm [5]. Apart from the purification process, metal ions could be supplemented to stabilize the structure of the aptamer during the sample preparation process. The aptamer selected against the Fc fragment of human IgG1 (hFc1) displays a typical structure fit to hFc1 that is stabilized by a calcium ion, ending up with the binding activity of the aptamer can be governed by calcium chelation and addition [43]. Given the methods utilized in the above studies, the addition of alkaline, acidic or metal ions into the buffer of the aptamer sample solution may enable the sample to be stable and to acquire a homogenous spectrum for structure determination.

## 5. Discussion

### 5.1. Structural Determination for Aptamer–Protein Complexes

Until now, we still do not have sufficient knowledge about the mechanism of DNA or RNA aptamers adapting structurally to bind proteins that are not physiologically DNA/RNA bounded proteins. A bunch of typical features have emerged with the investigation of more and more structural studies of aptamer–protein complexes. Aptamer surfaces adopt characteristic structural motifs to create a scaffold on which nucleotides that bind to the protein are precisely organized and exhibited. Protein surfaces present distinct structured interaction sites, or epitopes, that are recognized by aptamers and in most cases, the identical protein epitope can bind to aptamers with diverse sequences and potentially various structures [133,134]. The characterization of most aptamer–protein interactions has been narrowed to affinity or kinetic evaluation with few high-resolution structures of aptamer–protein complexes published [135]. Thus, there is a need to deeply investigate the degree of resemblances among the binding patterns, the preservation of intermolecular interactions and the structural heterogeneity of the aptamers.

The low affinity and stability of aptamers are big challenges in structure determination. Chemical modifications could improve the affinity and stability of the aptamer–protein interactions and, therefore, are helpful in structure determination. In turn, structure-guided modifications could further improve the affinity and stability of the aptamer–protein interaction and facilitate the translational development of aptamers. 

NMR technique was employed to calculate almost all the three-dimensional aptamer structures that have been investigated so far, as well as most aptamer–target binding complexes. Nonetheless, due to the overlap mapping of the complex NMR data, it is technically constrained to the resolution of the molecules that possess relatively small molecular weights [136]. From this perspective, X-ray crystallography is more suitable and powerful for investigating the aptamer–protein complexes. While the most common difficulties related to X-ray crystallography are the process of acquiring high-quality crystals, which relies on some restricting factors, including the purity and stability of the nucleic acid, the features and properties of the proteins to be crystallized, as well as the proportion between the proteins and aptamers, etc. [135]. An alternative and innovative method, cryogenic electron microscopy (cryo-EM), is employed to solve the three-dimensional structure of biological macromolecules under solvent conditions instead of crystals [137]. Whenever problems in obtaining high-quality crystals are too challenging to solve, cryo-EM may be substituted to calculate the structure of the complexes and to investigate the functional study.

### 5.2. In-Silico Analysis for Aptamer–Protein Binding

The availability of the 3D structure of the aptamer–target complex is essential for comprehending the structural mechanism underlying the interactions, thus performing molecular docking simulations is a very advantageous tactic. When it comes to the situation that we may not have access to the three-dimensional structure of the aptamer–macromolecular binding complexes, computational methods can be used to predict probable interaction sites, the most fitting binding mode and the binding forces that possibly exist between aptamers and targets, etc. Generally, the computational prediction approaches of aptamer have been proposed to carry out in two main categories: structure-based predictions and sequence-based predictions [138]. Structure-based techniques are widely utilized in computer-assisted drug design, which includes the secondary and tertiary structure prediction approaches for aptamers, molecular docking and molecular dynamic simulation methods for establishing aptamer-target binding models. In terms of predicting the binding capacities between targets and ligands in drug discovery, state-of-the-art machine/deep-learning models have witnessed abundant successes for years, thus potentially providing a robust and precise tool to predict the binding affinity between aptamers and targets just based on the sequences of the aptamers. In silico analysis would facilitate illustrating a more comprehensive interaction between the aptamer–protein complex that does not have the three-dimensional structure.

### 5.3. Different Application of Proteins Specific Aptamers

Interaction between aptamers and their targets could increase the stability of the protein and resistance to denaturation, which enable aptamers to become a new family of bio-recognition macromolecules for diagnostic and imaging applications. Detection of early-phase biomarkers have important roles in clinical diagnosis, and accurate biomarker detection during golden time could end up being lifesaving [139]. From this perspective, aptamers can be extremely important, including the aptamer generated for myoglobin in cardiovascular disease [140], ApoA1 for initial monitoring of hepatocellular carcinoma [141], CTAP III/NAP2 as indicator of lung cancer [142], retinol-binding protein-4 (RBP4), visceral adipose tissue-derived serpin (Vaspin) and nicotinamide phosphoribosyl transferase (Nampt/visfatin) adipokines for diagnosis of type 2 diabetes in the beginning of the diseases [143,144].

Another application of the aptamer is bio-imaging by conjugating the aptamer to a fluorophore, a QD, or other chemical elements, which is advantageous for magnetic resonance imaging (MRI) [145]. Since oligonucleotide moieties exist in the human body, adopting aptamers as imaging agents has the benefit of being non-toxic. Furthermore, the application of aptamers can enhance the certainty of the results acquired during diagnosis or clinical analysis because aptamers have high specificity to their target, precise targeting and swift diffusion through the blood circulation. According to these advantages, aptamers have been investigated as both imaging agents and single-protein imaging for cell imaging. Kim reported a work regarding C6 cell imaging using a Cy3-labeled AS1411 aptamer, which contained a chemical modification of 5-(N-benzylcarboxyamide)-2′-deoxyuridine (called 5′-BzdU) on a thymidine base [146]. In cancer cells, the AS1411 aptamer specifically targets the nucleolin transmembrane protein. Adding the group mentioned above strengthened the aptamer’s binding affinity to its target through a chemical modification. Compared to the original Cy3-labeled AS1411 aptamer, the cell imaging of the modified Cy3-labeled AS1411 aptamer was more efficient.

Taken together, despite the fact that aptamers have been promising therapeutic and diagnostic agents for decades in clinical trials, they are still not prevalently approved by the enormous drug market so far. Further illustration of the detailed interaction of the aptamer–target and modifications for affinity and activity improvement were essential for the progress on the promising application of the aptamer.

## Figures and Tables

**Figure 1 ijms-22-04093-f001:**
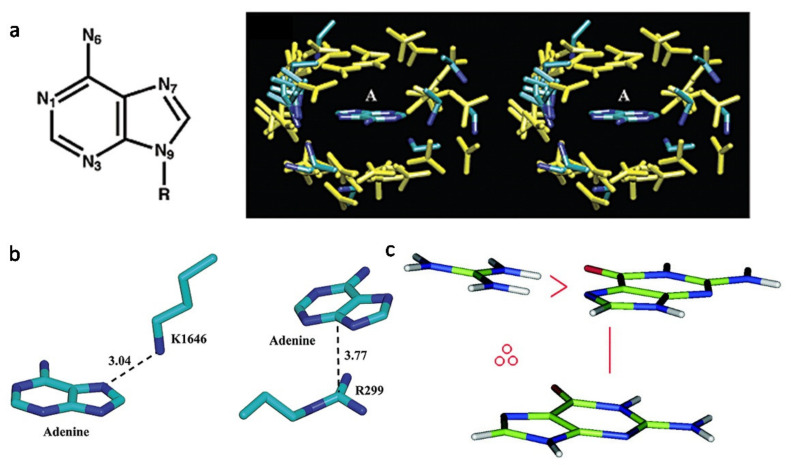
The interaction forces contribute to the aptamer–protein complex formation. (**a**) Pattern of cation–π interactions between charged residues and adenine. (**left**) Molecular structure of adenine. (**right**) A stereo diagram of adenine surrounded by positively charged residues in 68 adenine-binding proteins [68]. (**b**) Representative intermolecular interactions between adenine and positively charged residues. (**left**) Lys···Adenine pair in the TRP Ca-channel kinase domain (PDB ID: 1IA9); (**right**) Arg···Adenine pair in asparagine synthetase (PDB ID: 12AS). The atom−atom distance is indicated by a dashed line and is given in angstroms [68]. (**c**) A color picture to illustrate the H-bond/cation–π stair interaction motif. The depicted stair motif is of type G∴Arg∨G, where the symbol ∴ means cation–π interaction and ∨ H-bond. Cation–π interactions are defined geometrically by a distance and an angle criterion [69].

**Figure 2 ijms-22-04093-f002:**
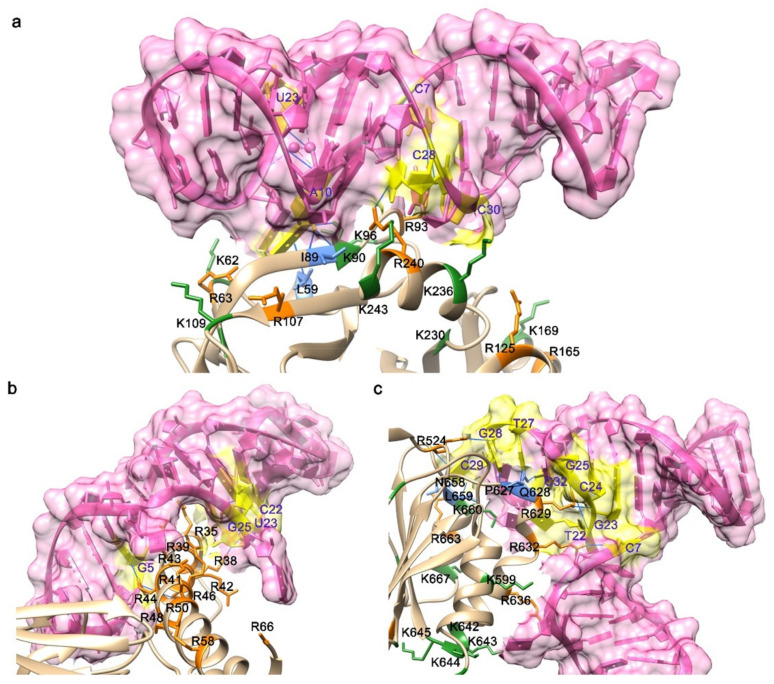
Interfaces of the aptamer–protein complexes exhibited common features of interaction. (**a**) The interface of the X-ray crystal structures of the RNA aptamer 11F7t bound to GD-FXaS195A (PDB ID: 5voe). (**b**) The interface of a potential anti-HIV-1 RNA aptamer in complex with Rev93: scFv (PDB ID: 6cf2). (**c**) The interface of the DNA aptamer ARC1172 in complex with VWF A1 domain (PDB ID: 3hxq). All aptamers were shown in pink, and the hydrogen bond was (blue solid lines) found between nucleotides (shown in yellow and labeled in blue) in the aptamer and residues on the proteins. The involved residues were mainly positive charged residues, including Lys (shown in green and labeled in black) and Arg (shown in orange and labeled in black). Other residues were non-charged residues (shown in green and labeled in black), including Leu, Ile, Pro and Asn.

**Figure 3 ijms-22-04093-f003:**
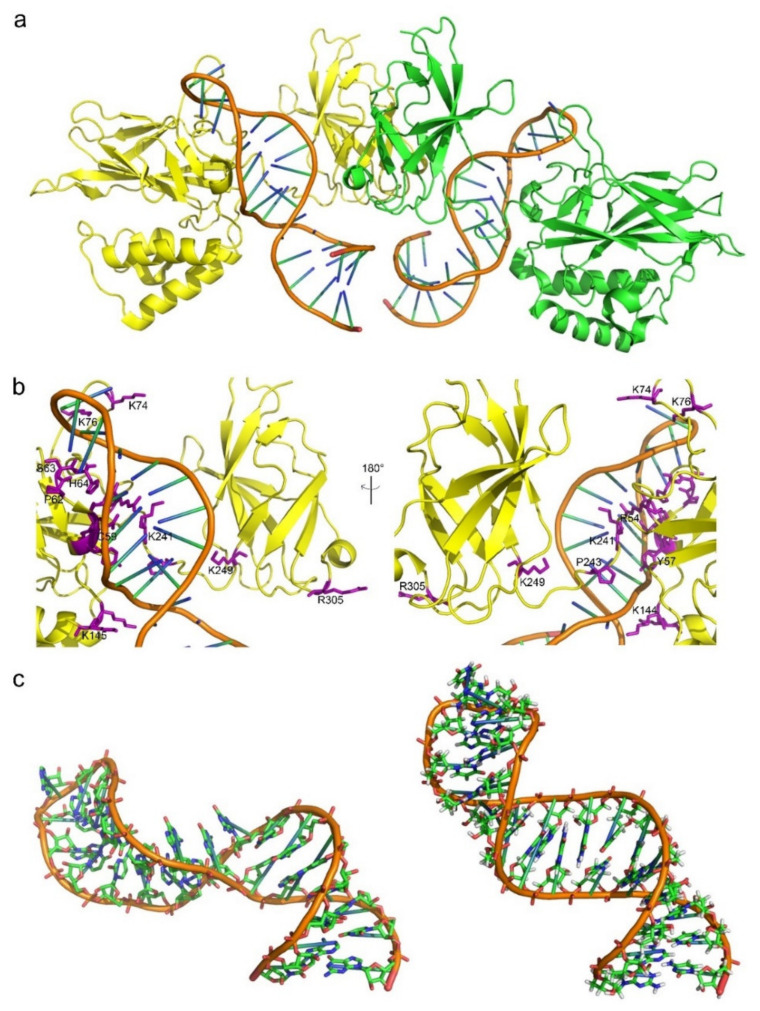
Representation of the state of the aptamer before and after bound to the target protein. (**a**) Anti-NF-κB RNA aptamer and crystal structure in complex with NF-κB p502 (PDB ID: 1ooa). In the crystal complex with NF-κB p502, one aptamer (orange) binds to each NF-κB subunit (yellow and green) of the dimer. (**b**) The interface of aptamer–protein complex showed the residues contributed to binding. The interaction residues were showed in purple, including mainly positive charged residues (Lys, Arg and His) and non-charged residues (Pro, Tyr and Cys). (**c**) Global structure of the bound (**left**) and free (**right**) anti-NF-kB RNA conformations. Measurement of the global axis curvature using CURVES 5.3 software [79] reveals 39.5 ± 10.08° and 97 ± 8.08° helical bend angles for the free (NMR) and protein-bound (crystal) states of the RNA, respectively [76].

**Figure 4 ijms-22-04093-f004:**
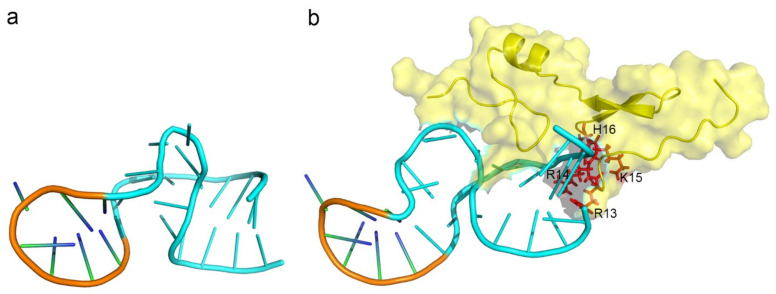
The model structure of Macugen (pegaptanib) and the induced fit-binding interaction with its target protein. (**a**) The structure model of the aptamer, pegaptanib, was shown according to the secondary structure calculated by NMR [131]. The conformation was the free form of the aptamer. (**b**) The interface of the aptamer in complex with the target protein, VEGF165 (PDB ID: 2VGH). The structure and the surface of VEGF165 was shown in yellow. The residues on pegaptanib that undergo a conformational change are shown in blue, while the other residues were shown in orange. The residues on VEGF165 that participated in the interaction of the complex were shown in red and labelled in black (R13, R14, K15, H16).

**Table 1 ijms-22-04093-t001:** Summary of three commonly used structural determination methods.

Method	Sample Requirement	Advantages	Limitations
NMR	Samples must be in solution	(1) Moelcules are studied in solution (2) Protein folding studies can be done by monitoring NMR spectra (3) Efficient in mapping interactions with other molecules	(1) The upper weight limit for NMR structure determination is ~30 kDa (2) Requires a protein sample be soluble in a high-concentrated solution (3) Overlapped spectrum
X-ray	Samples must be crystallized in a lattice structure	(1) Provides high-resolution information (2) Does not requie a protein be soluble in a high-concentrated solution (3) Applied to proteins or macromolecules in a wide molecular weight range	(1) Requires a protein crystal (2) Crystal contacts can distort protein structure (3) Not suitable with fairly flexible molecules
Cryo-EM	Sample is frozen in its native state	(1) High enough resolution (2) Does not require a protein crystal (3) Does not requie a protein be soluble in a high-concentrated solution	(1) Complex measurements and data analysis (2) Difficult to use for proteins with MW below 300 kDa (3) The technology still needs to be thoroughly tested by the scientific community

**Table 2 ijms-22-04093-t002:** The structural and affinity characteristics of aptamer–protein complexes.

PDB ID	Complex Type (Protein:mAptamer)	Polar Contacts (Number)	Interface Area (A^2^)	Dissociation Constant (nm)	Gibbs Free Energy Change (kJ/mol)	References
5mxf	Photorhabdus asymbiotica lectin (PHL): alpha-methyl fucoside	16	1165.2	1.4 ± 0.21 μm	−48.8	[32]
1u1y	MS2 coat protein: F5 RNA aptamer	42	3180.6	0.6 ± 0.3	−51.2	[33]
1exd	Glutamine aminoacyl tRNA synthetase: Glutamine tRNA	42	2599.3	0.3 ± 0.1	−35.9	[34]
1ooa	NF-kB(p50)2: RNA Aptamer	27	1343.1	5.4 ± 2.2	0.1	[35]
2b63	Complete RNA Polymerase II: RNA inhibitor	147	9311.5	33 ± 2	−91.8	[36]
4i7y	Human Alpha Thrombin: 27-mer DNA Aptamer	18	1079.5	0.7	−6.1	[37]
5ew1	Human thrombin: two DNA aptamers (HD22 and HD1-ΔT3)	21	1116.2	4.9 ± 1.6 51.8 ± 5.3	−10.0	[38,39]
4dii	Human alpha thrombin: thrombin binding DNA aptamer	32	1086.0	33	−9.4	[40,41]
3agv	Human IgG: RNA aptamer	13	1092.0	-	−12.8	[42,43]
3qlp	Human alpha thrombin: modified thrombin binding DNA aptamer	39	1404.0	25 ± 1	−9.5	[41,44]
3hxq	Von Willebrand Factor (VWF) A1 Domain: DNA Aptamer ARC1172	23	1070.8	2	−1.2	[45,46]
3uzs	G Protein-Coupled Receptor Kinase 2-Heterotrimeric G Protein Beta 1 and Gamma 2 Subunit Complex: C13.28 RNA Aptamer	13	2188.7	3.8 ± 1.2	−41.7	[47]
3uzt	G Protein-Coupled Receptor Kinase 2: C13.18 RNA Aptamer	8	1096.8	35 ± 5	−4.2	[47]
4hqx	Human PDGF-BB: Modified DNA aptamer (SOMAmer SL4)	24	1076.8	1.20	−14.4	[48]
4hqu	Human PDGF-BB: Modified DNA aptamer (SOMAmer SL5)	24	1078.6	0.02	−14.9	[48]
4lz4	Human thrombin: TBA (DNA) deletion mutant lacking thymine 3 nucleobase	43	1354.5	54.9	−10.6	[49]
5hto	Plasmodium Vivax LDH: DNA aptamer pL1	14	2105.1	16.8 ± 0.6	−37.3	[50]
4wb2	Mouse C5a complement anaphylatoxin: mirror-image L-RNA/L-DNA aptamer NOX-D20	14	420.6	0.02	−3.1	[51]
3zh2	Plasmodium falciparum lactate dehydrogenase: DNA aptamer	13	2106.3	42	−37.6	[52]
4pdb	Bacillus Anthracis Ribosomal Protein S8: RNA Aptamer	13	898.6	110 ± 30	−16.7	[5]
4ni7	Human interleukin 6: a modified DNA aptamer (SOMAmer SL1025)	3	336.3	0.20	3.0	[53]
4r8i	Chemokine CCL2: Mirror-image RNA Oligonucleotide Aptamer	10	807.3	1.4 ± 0.16	−8.8	[54]
4zbn	Nerve growth factor: non-helical DNA triplex aptamer	13	1418.9	0.21 ± 0.08	−25.2	[55]
5cmx	Human alpha thrombin: duplex/quadruplex DNA aptamer	32	1124.0	0.56	−9.0	[56]
5hrt	Mouse autotaxin: DNA aptamer	8	934.6	1.6	−11.7	[57]
5do4	Thrombin: RNA aptamer	31	1173.1	0.0081 ± 0.0002	−8.6	[58]
5uc6	IL-1 alpha: naphthyl-modified DNA aptamer	4	598.4	7.3	−1.1	[59]
6eo6	Human alpha-thrombin: modified 15-mer DNA aptamer	36	1064.1	1.00	−7.9	[60]
6sy4	TetR: TetR-binding RNA-aptamer K1	8	1615.2	5.6	−31.2	[61]
6rti	Human glutamate carboxypeptidase II: aptamer A9g	24	2414.7	-	−25.8	[62]
6z8w	Human alpha thrombin: thrombin binding aptamer variant (TBA-3G)	34	1066.1	9.8 ± 0.6	−8.1	[63]
7jtq	Human complement factor B: Slow off-rate modified aptamer	10	894.9	0.049	−3.1	[64]

**Table 3 ijms-22-04093-t003:** Advantages and disadvantages of different methods of detecting binding affinity.

	Principle	Advantages	Limitations
Isothermal titration calorimetry	Thermodynamics where contact between two molecules results in either exothermic or endothermic	(1) Precise detection of enthalpy change, stoichiometry and binding constant (2) Real-time and dynamic monitoring of the whole process of interaction (3) Application of interaction between proteins and ligands, polysaccharide, small compound	(1) Large quantity and high-quality demand high for samples; (2) The repeatability of identical sample reaction remains poor
Surface plasmon resonance	Changes in the local refractive index to detect adsorption onto microarrays	(1) Real-time and dynamic monitoring of the whole process of interaction (2) Maintain the structure and natural activity of the sample (3) The detection process is convenient and swift, with high sensitivity	(1) Sensitive to interference factors such as sample composition and temperature (2) Difficult to distinguish non-specific adsorption (3) Immobilization of one binding partner required
Atomic force microscopy	Detection between a ligand-functionalized AFM tip and a receptor-modified substrate	(1) Simple sample preparation; imaging under a lot of conditions (air, liquid, vacuum, physiological status) (2) Operated on one single molecule	(1) The between the tip and the sample could contaminate the tip, adsorb protein molecules or damage the surface of the sample (2) Salt crystallization of samples
Flow cytometry	The measurement of light scattered by particles and the fluorescence observed when the particles are passed in a stream through a laser beam.	(1) Tens of thousands of cells can be quickly examined, and the data gathered are processed by a computer (2) It provides quantifiable data from a sample	(1) Flow cytometry instrument is relatively expensive (2) Provides overwhelming information about the samples that may not be always necessary
Enzyme-linked oligonucleotide assay (ELONA)	A solid-phase type of enzyme immunoassay to detect the presence of a ligand in a liquid sample	(1) Aptamers can be easily labeled without significant effect on the affinity and specificity of the aptamer (2) Aptamers can provide a constant source of high and uniform quality detection reagent	(1) Sequence labeling required (2) Non-specific binding to the plate, which confuse the enrichment (3) Not suitable for small molecular targets
Surface-enhanced Raman spectroscopy	Enhances Raman scattering by molecules adsorbed on rough metal surfaces	(1) Requires relatively lower laser intensity and longer wavelengths (2) Rapid signal acquisition times	(1) Still requires interdisciplinary research effort to develop highly sensitive and reliable system
Microscale thermophoresis	Directed movement of molecules along temperature gradients, an effect termed thermophoresis	(1) Low sample consumption (2) Fast experimental procedure (3) Ability to perform measurements in complex samples, such as cell lysates (4) Possibility of labeling-free	(1) Cannot discern 2nd binding site or non-specific binding (2) Labeling with hydrophobic fluorophores required, which can alter binding profile
Bio-layer interferometry	A label-free method based on the real-time optical monitoring of biomolecular interactions	Highest throughput	(1) Immoblization of one binding partner required (2) Dissociation phases are imprecise due to analyte rebinding (no flow through system)

**Table 4 ijms-22-04093-t004:** Classification of chemical and structural modification strategies.

Modification Strategies	Categories
Chemical modifications	(1) Modifications on the inter-nucleotide linkages (2) Modifications on the nucleotide base
Strutural modifications	(1) Modifications on secondary structures (2) Modifications on the interactive nucleotides (3) Addition of structural groups

**Table 5 ijms-22-04093-t005:** Therapeutic aptamers currently in clinical trials.

Aptamer	Target	Therapy Area	Latest Clinical Trial Phase, Status
ARC1905	C5	Age-Related Macular Degeneration Juvenile Macular Degeneration (Stargardt Disease)	Phase 1, completed Phase 2, recruiting
Zimura	C5	Idiopathic Polypoidal Choroidal Vasculopathy	Phase 2, completed
Fovista	PDGF BB	Age-Related Macular Degeneration	Phase 1, terminated
E10030 plus Lucentis	PDGF	Age-Related Macular Degeneration	Phase 2, completed
EYE001	VEGF	Hippel–Lindau Disease Macular Degeneration Choroidal Neovascularization	Phase 2, completed Phase 2, completed Phase 3, completed
Macugen	VEGF165	Age-Related Macular Degeneration	Phase 3, completed
NU172	Thrombin	Heart disease	Phase 2, recruiting
68Ga-Sgc8	PTK7	Colorectal Cancer	Phase 1, unknown
ARC1779	vWF	Von Willebrand Disease Type-2b	Phase 2, completed
BT200	vWF	Von Willebrand Diseases Hemophilia A	Phase 2, completed
ApTOLL	TLR4	Stroke	Phase 1, completed
NOX-H94	Hepcidin peptide hormone	Anemia End-Stage Renal Disease	Phase 2, completed
NOX-E36	CCL2	Chronic Inflammatory Diseases Type 2 Diabetes Mellitus Systemic Lupus Erythematosus	Phase 1, completed
NOX-A12	CXCL12	Autologous Stem Cell Transplantation Hematopoietic Stem Cell Transplantation	Phase 1, completed Phase 1, completed
ARC19499	TFPI	Hemophilia	Phase 1, completed
AS1411	Nucleolin	Acute Myeloid Leukemia	Phase 2, terminated
AS1411-GNP	Nucleolin	Coronavirus Disease 2019 (COVID-19)	Phase 1, recruiting
ACTGRO-777	Nucleolin	Acute Myelocytic Leukemia Pancreatic Cancer	Phase 2, recruiting Phase 1, recruiting
RBM-007	FGF2	Exudative Age-Related Macular Degeneration	Phase 2, completed
BC-007	GPCR AAb	Dilated Cardiomyopathy Coronavirus Disease 2019 (COVID-19)	Phase 2, recruiting Phase 1, recruiting
REG1	Coagulation factor IXa	Coronary Artery Disease	Phase 2, completed

## Data Availability

No new data were created or analyzed in this study. Data sharing is not applicable to this article.

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
