# Peer review of "Structural Biology for the Molecular Insight between Aptamers and Target Proteins"

_ijms, 2021, doi:10.3390/ijms22084093_

Round 1

Reviewer 1 Report

In this manuscript, the authors summarized the methods for resolving structures of aptamer-protein complexes and for analyzing interactions between aptamers and target proteins. In addition, the approaches for determining the binding affinity between aptamers and target proteins as well as modification strategies for stabilizing the binding affinity of aptamers to target proteins were reviewed. Overall, the manuscript is well written and organized. Considering the importance of structural studies for aptamer-target interactions, this manuscript is worthy of publication and I suggest the acceptance of this manuscript after accommodating the following comments.

1) In section 4.1., the authors wrote “generally, aptamer have low affinity to their target proteins”, but it was contradicted with the following sentence in line 163 (quite a strong binding affinity between each pair). In addition, in line 428, the authors explained that the binding affinity of aptamer could match or even exceed that of antibody to its antigen. The authors need to clarify what the low affinity of aptamers to proteins means.

2) Because there are aptamers with the dissociation constant of pM level, the following sentence in line 162 should be modified: the constants fluctuate mildly around the levels of 10-100 nM.

3) The section 4.2, the authors need to include other methods including flow cytometry, ELONA, and SERS because they are also widely used to determine the binding affinity between aptamers and target proteins. In addition, the size of table 2 should be adjusted to fit into the manuscript file.

4) The authors briefly explained the computational methods to predict the potential binding sites between aptamers and targets. However, as the bioinformatics is in significant progress, it would be better to add the section for in-silico analysis. One example is doi.org/10.1016/j.jmgm.2021.107872

5) In the discussion section, it would be better to discuss how the structural analysis of interaction between aptamers and target proteins contribute to other research areas (e.g., diagnostics or imaging applications).

Reviewer 2 Report

Elucidation of aptamer-target interaction is a elusive target that still hampers the progress of aptamer field. Structural studies for aptamer-target interactions are still limited with various difficulties. In this manuscript, the methods for resolving structures of aptamer-protein complexes and analyzing the interactions between aptamers and target proteins were reviewed including the general features of the interacting nucleotides and residues involved in aptamer-protein interactions. Approaches for determining the binding affinity, modification strategies for better stability were also reviewed together with current challenges and perspectives.

Major comments

  1. How aptamers interact with their targets and how chemical and structural modifications affect the affinity and function of aptamers is an pressing issue that needs to be addressed in more depth, which will provide a basis for the development of viable aptamer in the clinic. Virtual analyses of aptamers in the clinical trials could be appended, including pre-formed and/or induced fit binding interactions in forming protein-aptamer complexes.
  2. The references cited should include more recent papers. Of the 71 references, the papers within the last 5 years are less than 25%.
  3. Cryo-EM and molecular docking part must be covered in more length and detail.

Minor points

  1. line 39. Almost 4 years have passed since the John Rossi paper has been published. The aptamers that are in clinical trials should be updated.
  2. line 240. ‘The interface of the of the DNA’ should be corrected.
  3. Table 3 can be modified and briefly summarized for better comparison and understanding.
  4. A table summarizing chemical and structural modifications of aptamers to enhance the stability and binding affinity of aptamers would help the reader.
  5. A table that compares NMR, X-ray crystallography, and Cryo-EM would improve the manuscript.

Reviewer 3 Report

This manuscript provides an overview  of some methods which can help to explore the interacting mechanism between aptamers and their targets. Author described some structure determination methods as well as  few biophysical methods and tried to summarize the general features of the interacting nucleotides and residues involved in the interactions between aptamers and proteins.

Modification strategies for stabilizing the binding affinity of aptamers to target proteins were also reviewed.

 The topic is interesting and important. The review of methods for determination of structures is almost complete but it concern only NMR technique and X-ray crystallography but cryogenic electron microscopy (cryo-EM) was only mentioned as an alternative and innovative method at the end of paper at discussion part. This methodology should be described wider and some examples should be given similar as for other methods, just because cryo-EM is very promising method and there were reported many models of structures of nucleic acid or complexes nucleic acids/protein using this methodology  (even if there are not reported examples of interaction of aptamer-target). Also others, but maybe less informative methods, like gel electrophoresis (e.g. for detection of G -quadruplex), circular dichroism, mass spectrometry that provide an understanding of the folding topology should be mentioned.

However a major missing in this paper concerns the part “Methods for determination of the binding affinity between aptamers and proteins”. Here, the authors listed only three methods and one of them is less popular (atomic force microscopy) although more common methods were not mentioned. Among these omitted methods are approaches which utilize fluorescent labeling, like fluorescence spectroscopy titrations or MST (microscale thermophoresis). Another known method for determination  of biomolecular interactions is bio-layer interferometry (BLI). All these methods should be described with examples. Additionally, it would be important to show the comparison of the binding affinity interactions measured by a few different methods (e.g. SPR, MST, BLI; see O’ Sullivan C.K. at al, International Journal of Molecular Sciences, 2021,  vol.22, p.1150) with explanation of differences (if any) between them.

Next part which have to be improved is “Modifications of aptamers to enhance the stability and binding affinity of aptamers”. The topic is very broad, so it should be structured in some way and may be numbered, e.g. (i) modifications of the sugar ring;  (ii) modifications of bases; (iii) modifications of the internucleotide phosphodiester linkage. Next typical examples for each group should be presented and next the case described by authors.

In concluding, the manuscript contain many remarkable information as well as interesting point of view but it should be filled by necessary knowledge at the topic which the authors wanted to comment.

Round 2

Reviewer 2 Report

Thanks for the revisions made according to the comments. But this reviewer still finds some concerns including the freshness of the review.

1. Section 4.3.3. does not cover the request made. The request was "Virtual analyses of aptamers in the clinical trials with respect to preformed and/or induced fit binding interactions in forming protein-aptamer complexes."

2. Another major concern is the clinical data that must be current. Please update the relevant parts (Section 1 and Section 4.3.3.) based on the data available at the time of writing. Section 4.3.3. deals with old references (2010 through 2018). Where are the references between 2019 and 2021? It seems that as of January, 2021 a total of 14 aptamers were registered in FDA database, including nine RNA and five DNA aptamers. Please recheck the current database, not relying only on the data that are outdated (2019, Reference 3). 

3. Please check the following for grammar and usage.

Line 19: which facing various difficulties.

Reviewer 3 Report

The authors responded to the comments and corrected the manuscript.

However I still have some remarks.

1) Table 1, row “NMR” and column  “Sample requirement” I would rather said that “Samples must be in solution “ instead of used: “Samples must be dissolved in water” because it is not true – it could be water with addition.

2) Table 1, row “X-ray”, column “Advantages”, point 3 – it is completely not true that “Applied to proteins with MW >200kDa”. I know crystal structures of protein with 13.7 kDa solved by X-ray. And crystallography is good technique to show structure of small molecules, e.g. nucleotides (MW~300 Da).

3) Table 3, method “Bio-layer interferometry “ column ” Limitations” authors listed “Immobilization of one binding partner required”. However the same limitation should be added for  method “Surface Plasmon Resonance”

4) Table 4 , method “Flow cytometry”, column “Principle” – it should be rather “ Cells (labeled or not) flow  in a liquid stream (sheath fluid), which carries and aligns the cells so that they pass single file through the light beam for sensing and the light scattered is characteristic to the cells and their components”. Or something similar but shorter.

In summary, this review is interesting and provides condensed knowledge at some aspect of aptamer research.
